# Incorporating BERT into Neural Machine Translation

**Jinhua Zhu**[1,*], **Yingce Xia**[2,*], **Lijun Wu**[3], **Di He**[4],
**Tao Qin**[2], **Wengang Zhou**[1], **Houqiang Li**[1], **Tie-Yan Liu**[2]

[1]CAS Key Laboratory of GIPAS, EEIS Department, University of Science and Technology of China;
[2]Microsoft Research;
[3]Sun Yat-sen University;
[4]Key Laboratory of Machine Perception (MOE), School of EECS, Peking University
[1]teslazhu@mail.ustc.edu.cn, {zhwg,lihq}@ustc.edu.cn
[2]yingce.xia@gmail.com, {taoqin,tyliu}@microsoft.com
[3]wulijun3@mail2.sysu.edu.cn [4]di_he@pku.edu.cn

## Abstract

The recently proposed BERT (Devlin et al., 2019) has shown great power on a variety of natural language understanding tasks, such as text classification, reading comprehension, etc. However, how to effectively apply BERT to neural machine translation (NMT) lacks enough exploration. While BERT is more commonly used as fine-tuning instead of contextual embedding for downstream language understanding tasks, in NMT, our preliminary exploration of using BERT as contextual embedding is better than using for fine-tuning. This motivates us to think how to better leverage BERT for NMT along this direction. We propose a new algorithm named BERT-fused model, in which we first use BERT to extract representations for an input sequence, and then the representations are fused with each layer of the encoder and decoder of the NMT model through attention mechanisms. We conduct experiments on supervised (including sentence-level and document-level translations), semi-supervised and unsupervised machine translation, and achieve state-of-the-art results on seven benchmark datasets. Our code is available at `https://github.com/bert-nmt/bert-nmt`.

## 1 Introduction

Recently, pre-training techniques, like ELMo (Peters et al., 2018), GPT/GPT-2 (Radford et al., 2018; 2019), BERT (Devlin et al., 2019), cross-lingual language model (briefly, XLM) (Lample & Conneau, 2019), XLNet (Yang et al., 2019b) and RoBERTa (Liu et al., 2019) have attracted more and more attention in machine learning and natural language processing communities. The models are first pre-trained on large amount of unlabeled data to capture rich representations of the input, and then applied to the downstream tasks by either providing context-aware embeddings of an input sequence (Peters et al., 2018), or initializing the parameters of the downstream model (Devlin et al., 2019) for fine-tuning. Such pre-training approaches lead to significant improvements on natural language understanding tasks. Among them, BERT is one of the most powerful techniques that inspires lots of variants like XLNet, XLM, RoBERTa and achieves state-of-the-art results for many language understanding tasks including reading comprehension, text classification, etc (Devlin et al., 2019).

Neural Machine Translation (NMT) aims to translate an input sequence from a source language to a target language. An NMT model usually consists of an encoder to map an input sequence to hidden representations, and a decoder to decode hidden representations to generate a sentence in the target language. Given that BERT has achieved great success in language understanding tasks, a question worthy studying is how to incorporate BERT to improve NMT. Due to the computation resource limitation, training a BERT model from scratch is unaffordable for many researchers. Thus, we focus on the setting of leveraging a pre-trained BERT model (instead of training a BERT model from scratch) for NMT.

---

*This work is conducted at Microsoft Research Asia. The first two authors contributed equally to this work.

Given that there is limited work leveraging BERT for NMT, our first attempt is to try two previous strategies: (1) using BERT to initialize downstream models and then fine-tuning the models, and (2) using BERT as context-aware embeddings for downstream models. For the first strategy, following Devlin et al. (2019), we initialize the encoder of an NMT model with a pre-trained BERT model, and then finetune the NMT model on the downstream datasets. Unfortunately, we did not observe significant improvement. Using a pre-trained XLM (Lample & Conneau, 2019) model, a variant of BERT for machine translation, to warm up an NMT model is another choice. XLM has been verified to be helpful for WMT'16 Romanian-to-English translation. But when applied to a language domain beyond the corpus for training XLM (such as IWSLT dataset (Cettolo et al., 2014), which is about spoken languages) or when large bilingual data is available for downstream tasks, no significant improvement is observed neither. For the second strategy, following the practice of (Peters et al., 2018), we use BERT to provide context-aware embeddings for the NMT model. We find that this strategy outperforms the first one (please refer to Section 3 for more details). This motivates us to go along this direction and design more effective algorithms.

We propose a new algorithm, *BERT-fused model*, in which we exploit the representation from BERT by feeding it into all layers rather than served as input embeddings only. We use the attention mechanism to adaptively control how each layer interacts with the representations, and deal with the case that BERT module and NMT module might use different word segmentation rules, resulting in different sequence (i.e., representation) lengths. Compared to standard NMT, in addition to BERT, there are two extra attention modules, the BERT-encoder attention and BERT-decoder attention. An input sequence is first transformed into representations processed by BERT. Then, by the BERT-encoder attention module, each NMT encoder layer interacts with the representations obtained from BERT and eventually outputs fused representations leveraging both BERT and the NMT encoder. The decoder works similarly and fuses BERT representations and NMT encoder representations.

We conduct 14 experiments on various NMT tasks to verify our approach, including supervised, semi-supervised and unsupervised settings. For supervised NMT, we work on five tasks of IWSLT datasets and two WMT datasets. Specifically, we achieve 36.11 BLEU score on IWSLT'14 German-to-English translation, setting a new record on this task. We also work on two document-level translations of IWSLT, and further boost the BLEU score of German-to-English translation to 36.69. On WMT'14 datasets, we achieve 30.75 BLEU score on English-to-German translation and 43.78 on English-to-French translation, significantly better over the baselines. For semi-supervised NMT, we boost BLEU scores of WMT'16 Romanian-to-English translation with back translation (Sennrich et al., 2016b), a classic semi-supervised algorithm, from 37.73 to 39.10, achieving the best result on this task. Finally, we verify our algorithm on unsupervised English↔French and unsupervised English↔Romanian translations and also achieve state-of-the-art results.

## 2 BACKGROUND AND RELATED WORK

We briefly introduce the background of NMT and review current pre-training techniques.

**NMT** aims to translate an input sentence from the source language to the target one. An NMT model usually consists of an encoder, a decoder and an attention module. The encoder maps the input sequence to hidden representations and the decoder maps the hidden representations to the target sequence. The attention module is first introduced by Bahdanau et al. (2015), which is used to better align source words and target words. The encoder and decoder can be specialized as LSTM (Hochreiter & Schmidhuber, 1997; Sutskever et al., 2014; Wu et al., 2016), CNN (Gehring et al., 2017) and Transformer (Vaswani et al., 2017). A Transformer layer consists of three sub-layers, a self-attention layer that processes sequential data taking the context of each timestep into consideration, an optional encoder-decoder attention layer that bridges the input sequence and target sequence which exists in decoder only, and a feed-forward layer for non-linear transformation. Transformer achieves the state-of-the-art results for NMT (Barrault et al., 2019). In this work, we will use Transformer as the basic architecture of our model.

**Pre-training** has a long history in machine learning and natural language processing (Erhan et al., 2009; 2010). Mikolov et al. (2013) and Pennington et al. (2014) proposed to use distributional representations (i.e., word embeddings) for individual words. Dai & Le (2015) proposed to train a language model or an auto-encoder with unlabeled data and then leveraged the obtained model to finetune downstream tasks. Pre-training has attracted more and more attention in recent years

and achieved great improvements when the data scale becomes large and deep neural networks are employed. ELMo was proposed in Peters et al. (2018) based on bidirectional LSTMs and its pre-trained models are fed into downstream tasks as context-aware inputs. In GPT (Radford et al., 2018), a Transformer based language model is pre-trained on unlabeled dataset and then finetuned on downstream tasks. BERT (Devlin et al., 2019) is one of the widely adopted pre-training approach for model initialization. The architecture of BERT is the encoder of Transformer (Vaswani et al., 2017). Two kinds of objective functions are used in BERT training: (1) *Masked language modeling (MLM)*, where $15\%$ words in a sentence are masked and BERT is trained to predict them with their surrounding words. (2) *Next sentence prediction (NSP)*: Another task of pre-training BERT is to predict whether two input sequences are adjacent. For this purpose, the training corpus consists of tuples (`[cls]`, input 1, `[sep]`, input 2, `[sep]`), with learnable special tokens `[cls]` to classify whether `input 1` and `input 2` are adjacent and `[sep]` to segment two sentences, and with probability 50%, the second input is replaced with a random input. Variants of BERT have been proposed: In XLM (Lample & Conneau, 2019), the model is pre-trained based on multiple languages and *NSP* task is removed; in RoBERTa (Liu et al., 2019), more unlabeled data is leveraged without *NSP* task neither; in XLNet (Yang et al., 2019b), a permutation based modeling is introduced.

## 3  A PRELIMINARY EXPLORATION

While a few pieces of work (Lample & Conneau, 2019; Song et al., 2019) design specific pre-training methods for NMT, they are time and resource consuming given that they need to pre-train large models from scratch using large-scale data, and even one model for each language pair. In this work, we focus on the setting of using a pre-trained BERT model. Detailed model download links can be found in Appendix D.

Considering that pre-trained models have been utilized in two different ways for other natural language tasks, it is straightforward to try them for NMT. Following previous practice, we make the following attempts.

(I) Use pre-trained models to initialize the NMT model. There are different implementations for this approach. (1) Following (Devlin et al., 2019), we initialize the encoder of an NMT model with a pre-trained BERT. (2) Following (Lample & Conneau, 2019), we initialize the encoder and/or decoder of an NMT model with XLM.

(II) Use pre-trained models as inputs to the NMT model. Inspired from (Peters et al., 2018), we feed the outputs of the last layer of BERT to an NMT model as its inputs.

We conduct experiments on the IWSLT'14 English→German translation, a widely adopted dataset for machine translation consisting of $160k$ labeled sentence pairs. We choose Transformer (Vaswani et al., 2017) as the basic model architecture with `transformer_iwslt_de_en` configuration (a six-layer model with 36.7M parameters). The translation quality is evaluated by BLEU (Papineni et al., 2002) score; the larger, the better. Both BERT$_{base}$ and XLM models are pre-trained and we get them from the Web. More details about the experimental settings are included in Appendix A.2.

Table 1: Preliminary explorations on IWSLT'14 English→German translation.

| Algorithm | BLEU score |
|---|---|
| Standard Transformer | 28.57 |
| Use BERT to initialize the encoder of NMT | 27.14 |
| Use XLM to initialize the encoder of NMT | 28.22 |
| Use XLM to initialize the decoder of NMT | 26.13 |
| Use XLM to initialize both the encoder and decoder of NMT | 28.99 |
| Leveraging the output of BERT as embeddings | 29.67 |

The results are shown in Table 1. We have several observations: (1) Using BERT to initialize the encoder of NMT can only achieve 27.14 BLEU score, which is even worse than standard Transformer without using BERT. That is, simply using BERT to warm up an NMT model is not a good choice. (2) Using XLM to initialize the encoder or decoder respectively, we get 28.22 or 26.13 BLEU score, which does not outperform the baseline. If both modules are initialized with XLM, the BLEU score

is boosted to 28.99, slightly outperforming the baseline. Although XLM achieved great success on WMT'16 Romanian-to-English, we get limited improvement here. Our conjecture is that the XLM model is pre-trained on news data, which is out-of-domain for IWSLT dataset mainly about spoken languages and thus, leading to limited improvement. (3) When using the output of BERT as context-aware embeddings of the encoder, we achieve 29.67 BLEU, much better than using pre-trained models for initialization. This shows that leveraging BERT as a feature provider is more effective in NMT. This motivates us to take one step further and study how to fully exploit such features provided by pre-trained BERT models.

## 4 ALGORITHM

In this section, we first define the necessary notations, then introduce our proposed BERT-fused model and finally provide discussions with existing works.

**Notations** Let $\mathcal{X}$ and $\mathcal{Y}$ denote the source language domain and target language domain respectively, which are the collections of sentences with the corresponding languages. For any sentence $x \in \mathcal{X}$ and $y \in \mathcal{Y}$, let $l_x$ and $l_y$ denote the number of units (e.g., words or sub-words) in $x$ and $y$. The $i$-th unit in $x/y$ is denoted as $x_i/y_i$. Denote the encoder, decoder and BERT as `Enc`, `Dec` and `BERT` respectively. For ease of reference, we call the encoder and decoder in our work as the *NMT module*. W.l.o.g., we assume both the encoder and decoder consists of $L$ layers. Let $\mathtt{attn}(q, K, V)$ denote the attention layer, where $q$, $K$ and $V$ indicate query, key and value respectively (Vaswani et al., 2017). We use the same feed-forward layer as that used in (Vaswani et al., 2017) and denote it as `FFN`. Mathematical formulations of the above layers are left at Appendix E.

### 4.1 BERT-FUSED MODEL

An illustration of our algorithm is shown in Figure 1. Any input $x \in \mathcal{X}$ is progressively processed by the BERT, encoder and decoder.

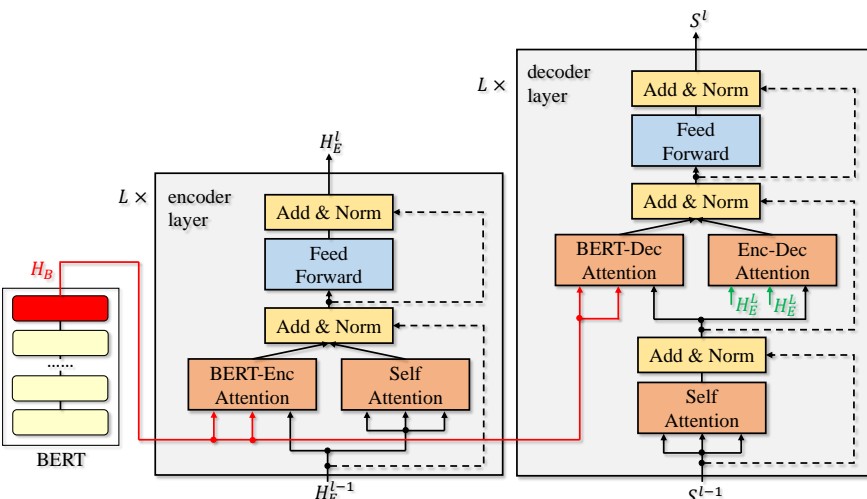

Figure 1: The architecture of BERT-fused model. The left and right figures represent the BERT, encoder and decoder respectively. Dash lines denote residual connections. $H_B$ (red part) and $H_E^L$ (green part) denote the output of the last layer from BERT and encoder.

*Step-1*: Given any input $x \in \mathcal{X}$, `BERT` first encodes it into representation $H_B = \mathtt{BERT}(x)$. $H_B$ is the output of the last layer in `BERT`. The $h_{B,i} \in H_B$ is the representation of the $i$-th wordpiece in $x$.

*Step-2*: Let $H_E^l$ denote the hidden representation of $l$-th layer in the encoder, and let $H_E^0$ denote word embedding of sequence $x$. Denote the $i$-th element in $H_E^l$ as $h_i^l$ for any $i \in [l_x]$. In the $l$-th

layer, $l \in [L]$,

$$\tilde{h}_i^l = \frac{1}{2}\big(\texttt{attn}_S(h_i^{l-1}, H_E^{l-1}, H_E^{l-1}) + \texttt{attn}_B(h_i^{l-1}, H_B, H_B)\big), \forall i \in [l_x], \tag{1}$$

where $\texttt{attn}_S$ and $\texttt{attn}_B$ are attention models (see Eqn.(6)) with different parameters. Then each $\tilde{h}_i^l$ is further processed by $\texttt{FFN}(\cdot)$ defined in Eqn.(7) and we get the output of the $l$-th layer: $H_E^l = (\texttt{FFN}(\tilde{h}_1^l), \cdots, \texttt{FFN}(\tilde{h}_{l_x}^l))$. The encoder will eventually output $H_E^L$ from the last layer.

*Step-3*: Let $S_{<t}^l$ denote the hidden state of $l$-th layer in the decoder preceding time step $t$, i.e., $S_{<t}^l = (s_1^l, \cdots, s_{t-1}^l)$. Note $s_1^0$ is a special token indicating the start of a sequence, and $s_t^0$ is the embedding of the predicted word at time-step $t-1$. At the $l$-th layer, we have

$$\hat{s}_t^l = \texttt{attn}_S(s_t^{l-1}, S_{<t+1}^{l-1}, S_{<t+1}^{l-1});$$
$$\tilde{s}_t^l = \frac{1}{2}\big(\texttt{attn}_B(\hat{s}_t^l, H_B, H_B) + \texttt{attn}_E(\hat{s}_t^l, H_E^L, H_E^L)\big), \ s_t^l = \texttt{FFN}(\tilde{s}_t^l). \tag{2}$$

The $\texttt{attn}_S$, $\texttt{attn}_B$ and $\texttt{attn}_E$ represent self-attention model, BERT-decoder attention model and encoder-decoder attention model respectively. Eqn.(2) iterates over layers and we can eventually obtain $s_t^L$. Finally $s_t^L$ is mapped via a linear transformation and softmax to get the $t$-th predicted word $\hat{y}_t$. The decoding process continues until meeting the end-of-sentence token.

In our framework, the output of BERT serves as an external sequence representation, and we use an attention model to incorporate it into the NMT model. This is a general way to leverage the pre-trained model regardless of the tokenization way.

## 4.2 DROP-NET TRICK

Inspired by dropout (Srivastava et al., 2014) and drop-path (Larsson et al., 2017), which can regularize the network training, we propose a drop-net trick to ensure that the features output by BERT and the conventional encoder are fully utilized. The drop-net will effect Eqn.(1) and Eqn.(2). Denote the drop-net rate as $p_{\text{net}} \in [0, 1]$. At each training iteration, for any layer $l$, we uniformly sample a random variable $U^l$ from $[0, 1]$, then all the $\tilde{h}_i^l$ in Eqn.(1) are calculated in the following way:

$$\tilde{h}_{i,\text{drop-net}}^l = \mathbb{I}\big(U^l < \frac{p_{\text{net}}}{2}\big) \cdot \texttt{attn}_S(h_i^{l-1}, H_E^{l-1}, H_E^{l-1}) + \mathbb{I}\big(U^l > 1 - \frac{p_{\text{net}}}{2}\big) \cdot \texttt{attn}_B(h_i^{l-1}, H_B, H_B)$$
$$+ \frac{1}{2}\mathbb{I}\big(\frac{p_{\text{net}}}{2} \le U^l \le 1 - \frac{p_{\text{net}}}{2}\big) \cdot \big(\texttt{attn}_S(h_i^{l-1}, H_E^{l-1}, H_E^{l-1}) + \texttt{attn}_B(h_i^{l-1}, H_B, H_B)\big), \tag{3}$$

where $\mathbb{I}(\cdot)$ is the indicator function. For any layer, with probability $p_{\text{net}}/2$, either the BERT-encoder attention or self-attention is used only; w.p. $(1 - p_{\text{net}})$, both the two attention models are used. For example, at a specific iteration, the first layer might uses $\texttt{attn}_S$ only while the second layer uses $\texttt{attn}_B$ only. During inference time, the expected output of each attention model is used, which is $\mathbb{E}_{U \sim \text{uniform}[0,1]}(\tilde{h}_{i,\text{drop-net}}^l)$. The expectation is exactly Eqn.(1).

Similarly, for training of the decoder, with the drop-net trick, we have

$$\tilde{s}_{t,\text{drop-net}}^l = \mathbb{I}(U^l < \frac{p_{\text{net}}}{2}) \cdot \texttt{attn}_B(\hat{s}_t^l, H_B, H_B) + \mathbb{I}(U^l > 1 - \frac{p_{\text{net}}}{2}) \cdot \texttt{attn}_E(\hat{s}_t^l, H_E^L, H_E^L)$$
$$+ \frac{1}{2}\mathbb{I}(\frac{p_{\text{net}}}{2} \le U^l \le 1 - \frac{p_{\text{net}}}{2}) \cdot \big(\texttt{attn}_B(\hat{s}_t^l, H_B, H_B) + \texttt{attn}_E(\hat{s}_t^l, H_E^L, H_E^L)\big). \tag{4}$$

For inference, it is calculated in the same way as Eqn.(2). Using this technique can prevent network from overfitting (see the second part of Section 6 for more details).

## 4.3 DISCUSSION

*Comparison with ELMo* As introduced in Section 2, ELMo (Peters et al., 2018) provides a context-aware embeddings for the encoder in order to capture richer information of the input sequence. Our approach is a more effective way of leveraging the features from the pre-trained model: (1) The output features of the pre-trained model are fused in all layers of the NMT module, ensuring the well-pre-trained features are fully exploited; (2) We use the attention model to bridge the NMT module and the pre-trained features of BERT, in which the NMT module can adaptively determine how to leverage the features from BERT.

*Limitations* We are aware that our approach has several limitations. (1) Additional storage cost: our approach leverages a BERT model, which results in additional storage cost. However, considering the BLEU improvement and the fact that we do not need additional training of BERT, we believe that the additional storage is acceptable. (2) Additional inference time: We use BERT to encode the input sequence, which takes about $45\%$ additional time (see Appendix C for details). We will leave the improvement of the above two limitations as future work.

## 5 APPLICATION TO SUPERVISED NMT AND SEMI-SUPERVISED NMT

We first verify our BERT-fused model on the supervised setting, including low-resource and rich-resource scenarios. Then we conduct experiments on document-level translation to verify our approach. Finally, we combine BERT-fused model with back translation (Sennrich et al., 2016b) to verify the effectiveness of our method on semi-supervised NMT.

### 5.1 SETTINGS

**Dataset** For the low-resource scenario, we choose IWSLT'14 English↔German (En↔De), English→Spanish (En→Es), IWSLT'17 English→French (En→Fr) and English→Chinese (En→Zh) translation. There are $160k$, $183k$, $236k$, $235k$ bilingual sentence pairs for En↔De, En→Es, En→Fr and En→Zh tasks. Following the common practice (Edunov et al., 2018), for En↔De, we lowercase all words. All sentences are preprocessed by BPE (Sennrich et al., 2016c). The model configuration is `transformer_iwslt_de_en`, representing a six-layer model with embedding size $512$ and FFN layer dimension $1024$. For the rich-resource scenario, we work on WMT'14 En→De and En→Fr, whose corpus sizes are $4.5M$ and $36M$ respectively. We concatenate newstest2012 and newstest2013 as the validation set and use newstest2014 as the test set. The model configuration is `transformer_big`, another six-layer network with embedding size $1024$ and FFN layer dimension $4096$. More details about data and model are left in Appendix A.1.

We choose $BERT_{base}$ for IWSLT tasks and $BERT_{large}$ for WMT tasks, which can ensure that the dimension of the BERT and NMT model almost match. The BERT models are fixed during training. Detailed BERT information for each task is in Appendix D. The drop-net rate $p_{net}$ is set as $1.0$.

**Training Strategy** We first train an NMT model until convergence, then initialize the encoder and decoder of the BERT-fused model with the obtained model. The BERT-encoder attention and BERT-decoder attention are randomly initialized. Experiments on IWSLT and WMT tasks are conducted on 1 and 8 M40 GPUs respectively. The batchsize is $4k$ tokens per GPU. Following (Ott et al., 2018), for WMT tasks, we accumulate the gradient for 16 iterations and then update to simulate a 128-GPU environment. It takes 1, 8 and 14 days to obtain the pre-trained NMT models, and additional 1, 7 and 10 days to finish the whole training process. The optimization algorithm is Adam (Kingma & Ba, 2014) with initial learning rate $0.0005$ and `inverse_sqrt` learning rate scheduler (Vaswani et al., 2017). For WMT'14 En→De, we use beam search with width $4$ and length penalty $0.6$ for inference following (Vaswani et al., 2017). For other tasks, we use width $5$ and length penalty $1.0$.

**Evaluation** We use `multi-bleu.perl` to evaluate IWSLT'14 En↔De and WMT translation tasks for fair comparison with previous work. For the remaining tasks, we use a more advance implementation of BLEU score, `sacreBLEU` for evaluation. Script urls are in Appendix A.1.

### 5.2 RESULTS

The results of IWSLT translation tasks are reported in Table 2. We implemented standard Transformer as baseline. Our proposed BERT-fused model can improve the BLEU scores of the five tasks by 1.88, 1.47, 2.4, 1.9 and 2.8 points respectively, demonstrating the effectiveness of our method. The consistent improvements on various tasks shows that our method works well for low-resource translations. We achieved state-of-the-art results on IWSLT'14 De→En translation, a widely investigated baseline in ma-

Table 2: BLEU of all IWSLT tasks.

|  | Transformer | BERT-fused |
|---|---|---|
| En→De | 28.57 | 30.45 |
| De→En | 34.64 | 36.11 |
| En→Es | 39.0 | 41.4 |
| En→Zh | 26.3 | 28.2 |
| En→Fr | 35.9 | 38.7 |

chine translation. The comparison with previous methods are shown in Appendix B.4 due to space limitation.

The results of WMT'14 En→De and En→Fr are shown in Table 3. Our reproduced Transformer matches the results reported in Ott et al. (2018), and we can see that our BERT-fused model can improve these two numbers to 30.75 and 43.78, achieving 1.63 and 0.82 points improvement. Our approach also outperforms the well-designed model DynamicConv (Wu et al., 2019) and a model obtained through neural architecture search (So et al., 2019).

Table 3: BLEU scores of WMT'14 translation.

| Algorithm | En→De | En→Fr |
|---|---|---|
| DynamicConv (Wu et al., 2019) | 29.7 | 43.2 |
| Evolved Transformer (So et al., 2019) | 29.8 | 41.3 |
| Transformer + Large Batch (Ott et al., 2018) | 29.3 | 43.0 |
| Our Reproduced Transformer | 29.12 | 42.96 |
| Our BERT-fused model | 30.75 | 43.78 |

## 5.3 TRANSLATION WITH DOCUMENT-LEVEL CONTEXTUAL INFORMATION

BERT is able to capture the relation between two sentences, since the *next sentence prediction (NSP)* task is to predict whether two sentences are adjacent. We can leverage this property to improve translation with document-level contextual information (Miculicich et al., 2018), which is briefly denoted as document-level translation. The inputs are a couple of sentences extracted from a paragraph/document, $x_1^d, x_2^d, \cdots, x_T^d$, where the $T$ $x$'s are contextually correlated. We want to translate them into target language by considering the contextual information.

**Algorithm** In our implementation, to translate a sentence $x$ to target domain, we leverage the contextual information by taking both $x$ and its preceding sentence $x_{\text{prev}}$ as inputs. $x$ is fed into `Enc`, which is the same as sentence-level translation. For the input of `BERT`, it is the concatenation of two sequences: (`[cls]`, $x_{\text{prev}}$, `[sep]`, $x$, `[sep]`), where both `[cls]` and `[sep]` are special tokens of `BERT`.

**Setting** We use IWSLT'14 En↔De dataset as introduced in Section 5.1. The data is a collection of TED talks, where each talk consists of several sequences. We can extract the adjacent sentences for training, validation and test sets. The training strategy, hyperparameter selection and evaluation metric are the same for sentence-level translation.

**Baselines** We use two baselines here. (1) To demonstrate how `BERT` works in our model, we replace `BERT` by a Transformer with configuration `transformer_iwslt_de_en`, which is randomly initialized and jointly trained. (2) Another baseline is proposed by Miculicich et al. (2018), where multiple preceding sentences in a document are leveraged using a hierarchical attention network.

Table 4: BLEU of document-level translation.

| | En→De | De→En |
|---|---|---|
| Sentence-level | 28.57 | 34.64 |
| Our Document-level | 28.90 | 34.95 |
| Miculicich et al. (2018) | 27.94 | 33.97 |
| Sentence-level + BERT | 30.45 | 36.11 |
| Document-level + BERT | 31.02 | 36.69 |

**Results** The results are shown in Table 4. We can see that introducing contextual information from an additional encoder can boost the sentence-level baselines, but the improvement is limited (0.33 for En→De and 0.31 for De→En). For Miculicich et al. (2018), the best results we obtain are 27.94 and 33.97 respectively, which are worse than the sentence-level baselines. Combining BERT-fused model and document-level information, we can eventually achieve 31.02 for En→De and 36.69 for De→En. We perform significant test[1] between sentence-level and document-level translation. Our document-level BERT-fused model significantly outperforms sentence-level baseline with $p$-value less than 0.01. This shows that our approach not only works for sentence-level translation, but can also be generalized to document-level translation.

---

[1]`https://github.com/moses-smt/mosesdecoder/blob/master/scripts/analysis/bootstrap-hypothesis-difference-significance.pl`

## 5.4 APPLICATION TO SEMI-SUPERVISED NMT

We work on WMT'16 Romanian→English (Ro→En) translation to verify whether our approach can still make improvement over back translation (Sennrich et al., 2016b), the standard and powerful semi-supervised way to leverage monolingual data in NMT.

The number of bilingual sentence pairs for Ro→En is $0.6M$. Sennrich et al. (2016a) provided $2M$ back translated data[2]. We use newsdev2016 as validation set and newstest2016 as test set. Sentences were encoded using BPE with a shared source-target vocabulary of about $32k$ tokens. We use `transformer_big` configuration. Considering there is no Romanian BERT, we use the cased multilingual BERT (please refer to Appendix D) to encode inputs. The drop-net rate $p_{net}$ is set as $1.0$. The translation quality is evaluated by `multi-bleu.perl`.

The results are shown in Table 5. The Transformer baseline achieves 33.12 BLEU score. With back-translation, the performance is boosted to 37.73. We use the model obtained with back-translation to initialize BERT-fused model, and eventually reach 39.10 BLEU. Such a score surpasses the previous best result 38.5 achieved by XLM (Lample & Conneau, 2019) and sets a new record. This demonstrates that our proposed approach is effective and can still achieve improvement over strong baselines.

Table 5: BLEU scores of WMT'16 Ro→En.

| Methods | BLEU |
|---|---|
| Sennrich et al. (2016a) | 33.9 |
| XLM (Lample & Conneau, 2019) | 38.5 |
| Standard Transformer | 33.12 |
| + back translation | 37.73 |
| + BERT-fused model | 39.10 |

## 6 ABLATION STUDY

We conduct two groups of ablation studies on IWSLT'14 En→De translation to better understand our model.

Table 6: Ablation study on IWSLT'14 En→De.

| | |
|---|---|
| Standard Transformer | 28.57 |
| BERT-fused model | 30.45 |
| Randomly initialize encoder/decoder of BERT-fused model | 27.03 |
| Jointly tune BERT and encoder/decoder of BERT-fused model | 28.87 |
| Feed BERT feature into all layers without attention | 29.61 |
| Replace BERT output with random vectors | 28.91 |
| Replace BERT with the encoder of another Transformer model | 28.99 |
| Remove BERT-encoder attention | 29.87 |
| Remove BERT-decoder attention | 29.90 |

**Study for training strategy and network architecture**

We conduct ablation study to investigate the performance of each component of our model and training strategy. Results are reported in Table 6:

**(1)** We randomly initialize the NMT module (i.e., encoder and decoder) of BERT-fused model instead of using a warm-start one as introduced in the training strategy of Section 5.1. In this way, we can only achieve 27.03 BLEU score, which cannot catch up with the baseline. We also jointly train BERT model with the NMT module. Although it can also boost the baseline from 28.57 to 28.87, it is not as good as fixing the BERT part, whose BLEU is 30.45.

**(2)** We feed the output of BERT into all layers of the encoder without attention models. That is, the Eqn.(1) is revised to $\tilde{h}_i^l = \frac{1}{2}\left(\text{attn}_S(h_i^{l-1}, H_E^{l-1}, H_E^{l-1}) + W_B^l h_i^{l-1})\right)$, where $W_B^l$ is learnable. In this case, the encoder and BERT have to share the same vocabulary. The BLEU score is 29.61, which is better than the standard Transformer but slightly worse than leveraging the output of BERT

---

[2]Data at `http://data.statmt.org/rsennrich/wmt16_backtranslations/ro-en/`.

as embedding. This shows that the output of BERT should not be fused into each layer directly, and using the attention model to bridge the relation is better than using simple transformation. More results on different languages are included in Appendix B.3. To illustrate the effectiveness of our method, we choose another two kinds of ways to encode the input sequence rather than using BERT: (1) Using a fixed and randomly initialized embedding; (2) Using the encoder from another NMT model. Their BLEU scores are 28.91 and 28.99 respectively, indicating that the BERT pre-trained on large amount of unlabeled data can provide more helpful features to NMT.

**(3)** To verify where the output of BERT should be connected to, we remove the BERT-encoder attention (i.e., $\mathtt{attn}_B$ in Eqn.(1)) and the BERT-decoder attention (i.e,, $\mathtt{attn}_B$ in Eqn.(2)) respectively. Correspondingly, the BLEU score drops from 30.45 to 29.87 and 29.90. This indicates that the output of BERT should be leveraged by both encoder and decoder to achieve better performances. At last, considering that there are two stacked encoders in our model, we also choose ensemble models and deeper NMT models as baselines. Our approach outperforms the above baselines. The results are left in Appendix B.2 due to space limitation.

### Study on drop-net

To investigate the effect of drop-net, we conduct experiments on IWSLT'14 En→De dataset with different drop-net probability, $p_{net} \in \{0, 0.2, 0.4, 0.6, 0.8, 1.0\}$. The results are shown in Figure 2. As can been seen, although larger $p_{net}$ leads to larger training loss, it leads to smaller validation loss and so better BLUE scores. This shows that the drop-net trick can indeed improve the generalization ability of our model. We fix $p_{net} = 1.0$ in other experiments unless specially specified.

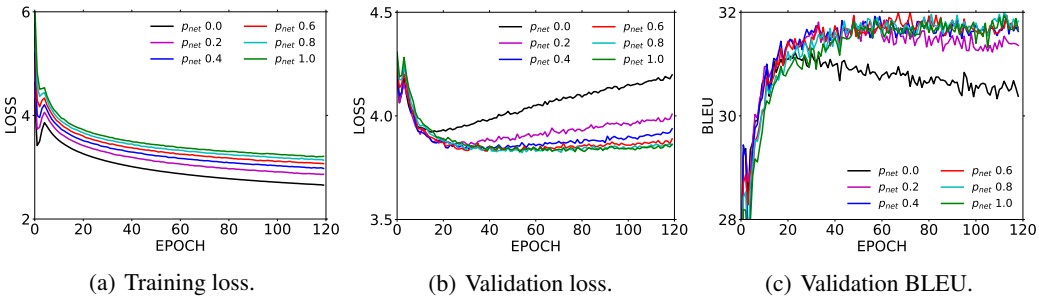

(a) Training loss.      (b) Validation loss.      (c) Validation BLEU.

Figure 2: Training/validation curves with different $p_{net}$'s.

## 7 APPLICATION TO UNSUPERVISED NMT

We work on unsupervised En↔Fr and En↔Ro translation. The data processing, architecture selection and training strategy is the same as Lample & Conneau (2019).

**Settings** For En↔Fr, we use $190M$ monolingual English sentences and $62M$ monolingual French sentences from WMT News Crawl datasets, which is the same as that used in (Song et al., 2019).[3] For unsupervised En↔Ro translation, we use $50M$ English sentences from News Crawl (sampled from the data for En→Fr) and collect $2.9M$ sentences for Romanian by concatenating News Crawl data sets and WMT'16 Romanian monolingual data following Lample et al. (2018). The data is preprocessed in the same way as Lample & Conneau (2019).

We use the same model configuration as Lample & Conneau (2019), with details in Appendix A.3. The `BERT` is the pre-trained XLM model (see Appendix D). We first train an unsupervised NMT model following Lample & Conneau (2019) until convergence. Then we initialize our BERT-fused model with the obtained model and continue training. We train models on 8 M40 GPUs, and the batchsize is 2000 tokens per GPU. We use the same optimization hyper-parameters as that described in Lample & Conneau (2019).

---

[3]Data source: `https://modelrelease.blob.core.windows.net/mass/en-fr.tar.gz`.

Table 7: BLEU scores of unsupervised NMT.

|  | En→Fr | Fr→En | En→Ro | Ro→En |
|---|---|---|---|---|
| Lample et al. (2018) | 27.6 | 27.7 | 25.1 | 23.9 |
| XLM (Lample & Conneau, 2019) | 33.4 | 33.3 | 33.3 | 31.8 |
| MASS (Song et al., 2019) | 37.50 | 34.90 | 35.20 | 33.10 |
| Our BERT-fused model | 38.27 | 35.62 | 36.02 | 33.20 |

**Results** The results of unsupervised NMT are shown in Table 7. With our proposed BERT-fused model, we can achieve 38.27, 35.62, 36.02 and 33.20 BLEU scores on the four tasks, setting state-of-the-art results on these tasks. Therefore, our BERT-fused model also benefits unsupervised NMT.

## 8 CONCLUSION AND FUTURE WORK

In this work, we propose an effective approach, BERT-fused model, to combine BERT and NMT, where the BERT is leveraged by the encoder and decoder through attention models. Experiments on supervised NMT (including sentence-level and document-level translations), semi-supervised NMT and unsupervised NMT demonstrate the effectiveness of our method.

For future work, there are many interesting directions. First, we will study how to speed up inference time. Second, we can apply such an algorithm to more applications, like questioning and answering. Third, how to compress BERT-fused model into a light version is another topic. There are some contemporary works leveraging knowledge distillation to combine pre-trained models with NMT (Yang et al., 2019a; Chen et al., 2019), which is a direction to explore.

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

## A  EXPERIMENT SETUP

### A.1  IWSLT'14 & WMT'14 SETTINGS

We mainly follow the scripts below to preprocess the data: `https://github.com/pytorch/fairseq/tree/master/examples/translation`.

**Dataset** For the low-resource scenario, we choose IWSLT'14 English↔German (En↔De), English→Spanish (En→Es), IWSLT'17 English→French (En→Fr) and English→Chinese (En→Zh) translation. There are $160k$, $183k$, $236k$, $235k$ bilingual sentence pairs for En↔De, En→Es, En→Fr and En→Zh tasks. Following the common practice (Edunov et al., 2018), for En↔De, we lowercase all words, split $7k$ sentence pairs from the training dataset for validation and concatenate *dev2010*, *dev2012*, *tst2010*, *tst2011*, *tst2012* as the test set. For other tasks, we do not lowercase the words and use the official validation/test sets of the corresponding years.

For rich-resource scenario, we work on WMT'14 En→De and En→Fr, whose corpus sizes are $4.5M$ and $36M$ respectively. We concatenate newstest2012 and newstest2013 as the validation set and use newstest2014 as the test set.

We apply BPE (Sennrich et al., 2016c) to split words into sub-units. The numbers of BPE merge operation for IWSLT tasks, WMT'14 En→De and En→Fr are $10k$, $32k$ and $40k$ respectively. We merge the source and target language sentences for all tasks to build the vocabulary except En→Zh.

**Model Configuration** For IWSLT tasks, we use the `transformer_iwslt_de_en` setting with dropout ratio 0.3. In this setting, the embedding dimension, FFN layer dimension and number of layers are 512, 1024 and 6. For WMT'14 En→De and En→Fr, we use `transformer_big` setting (short for `transformer_vaswani_wmt_en_de_big`) with dropout 0.3 and 0.1 respectively. In this setting, the aforementioned three parameters are 1024, 4096 and 6 respectively.

**Evaluation** We use `multi-bleu.perl`[4] to evaluate IWSLT'14 En↔De and WMT translation tasks for fair comparison with previous work. For the remaining tasks, we use a more advance implementation of BLEU score, detokenized sacreBLEU for evaluation[5].

### A.2  DETAILED EXPERIMENT SETTING IN SECTION 3

The IWSLT'14 English-to-German data and model configuration is introduced in Section A.1.

For the training stategy, we use Adam (Kingma & Ba, 2014) to optimize the network with $\beta_1 = 0.9$, $\beta_2 = 0.98$ and weight-decay $= 0.0001$. The learning rate scheduler is `inverse_sqrt`, where `warmup-init-lr` $= 10^{-7}$, `warmup-updates` $= 4000$ and `max-lr` $= 0.0005$.

### A.3  DETAILED MODEL CONFIGURATION IN UNSUPERVISED NMT

We leverage one Transformer model with GELU activation function to work on translations of two directions, where each language is associated with a language tag. The embedding dimension, FFN layer dimension and number of layer are 1024, 4096 and 6. The BERT is initialized by the pre-trained XLM model provided by (Lample & Conneau, 2019).

## B  MORE EXPERIMENT RESULTS

### B.1  MORE RESULTS ON PRELIMINARY EXPLORATION OF LEVERAGING BERT

We use XLM to initialize the model for WMT'14 English→German translation task, whose training corpus is relative large. We eventually obtain 28.09 after 90 epochs, which is still underperform the baseline, 29.12 as we got. Similar problem is also reported in `https://github.com/facebookresearch/XLM/issues/32`. We leave the improvement of supervised NMT with XLM as future work.

---

[4] `https://github.com/moses-smt/mosesdecoder/blob/master/scripts/generic/multi-bleu.perl`

[5] `https://github.com/mjpost/sacreBLEU`.

B.2 MORE ABLATION STUDY

**Part I: A different way to deal with multiple attention models**

Junczys-Dowmunt & Grundkiewicz (2018) proposed a new way to handle multiple attention models. Instead of using Eqn.(2), the input is processed by self-attention, encoder-decoder attention and BERT-decoder attention sequentially. Formally,

$$\hat{s}_t^l = \texttt{attn}_S(s_t^{l-1}, S_{<t+1}^{l-1}, S_{<t+1}^{l-1});$$
$$\bar{s}_t^l = \texttt{attn}_E(\hat{s}_t^l, H_E^L, H_E^L);$$
$$\tilde{s}_t^l = \texttt{attn}_B(\bar{s}_t^l, H_B, H_B);$$
$$s_t^l = \texttt{FFN}(\tilde{s}_t^l). \tag{5}$$

The BLEU score is 29.35 for this setting, not as good as our proposed method.

**Part II: More results on IWSLT'14 En→De translation**

Since our BERT-fused model contains two stacked encoders, we carry out two groups of additional baselines:

(1) Considering that stacking the BERT and encoder can be seen as a deeper model, we also train another two NMT models with deeper encoders, one with 18 layers (since BERT$_{base}$ consists of 12 layers) and the other with 12 layers (which achieved best validation performance ranging from 6 to 18 layers).

(2) We also compare the results of our approach with ensemble methods. To get an $M$-model ensemble, we independently train $M$ models with different random seeds ($M \in \mathbb{Z}_+$). We ensemble both standard Transformers and our BERT-fused models, which are denoted as $M$-model ensemble (standard) and $M$-model ensemble (BERT-fused) respectively. Please note that when we aggregate multiple BERT-fused models, we only need to store one replica of the BERT model because the BERT part is not optimized.

Table 8: More ablation study on IWSLT'14 En→De.

| Algorithm | BLEU |
|---|---|
| Standard Transformer | 28.57 |
| BERT-fused model | 30.45 |
| 12-layer encoder | 29.27 |
| 18-layer encoder | 28.92 |
| 2-model ensemble (standard) | 29.71 |
| 3-model ensemble (standard) | 30.08 |
| 4-model ensemble (standard) | 30.18 |
| 2-model ensemble (BERT-fused) | 31.09 |
| 3-model ensemble (BERT-fused) | 31.45 |
| 4-model ensemble (BERT-fused) | 31.85 |

The results are shown in Table 8. We have the following observations:

1. Adding more layers can indeed boost the baseline, but still not as good as BERT-fused model. According to our experiments, when increasing the number of layers to 12, we achieve the best BLEU score, 29.27.

2. We also compare our results to ensemble methods. Indeed, ensemble significantly boost the baseline by more than one point. However, even if using ensemble of four models, the BLEU score is still lower than our BERT-fused model (30.18 v.s. 30.45), which shows the effectiveness of our method.

We want to point out that our method is intrinsically different from ensemble. Ensemble approaches usually refer to "independently" train several different models for the same task, and then aggregate

the output of each model to get the eventually task. In BERT-fused model, although we include a pre-trained BERT into our model, there is still only one model serving for the translation task.

In this sense, we can also combine our BERT-fused model with ensemble. Our approach benefits from ensemble too. When ensembling two models, we can achieve 31.09 BLEU score. When adding the number of models to four, we eventually achieve 31.85 BLEU score, which is 1.67 point improvement over the ensemble of standard Transformer.

**Part III: More results on IWSLT'14 De→En translation**

We report the ensemble results on IWSLT'14 De→En translation in Table 9. We can get similar conclusion compared to that of IWSLT'14 En→De.

Table 9: More ablation study on IWSLT'14 De→En.

| Algorithm | BLEU |
|---|---|
| Standard Transformer | 34.67 |
| BERT-fused model | 36.11 |
| 2-model ensemble (standard) | 35.92 |
| 3-model ensemble (standard) | 36.40 |
| 4-model ensemble (standard) | 36.54 |
| 2-model ensemble (BERT-fused) | 37.42 |
| 3-model ensemble (BERT-fused) | 37.70 |
| 4-model ensemble (BERT-fused) | 37.71 |

## B.3 More results on feeding BERT output to NMT module

The ablation study on more languages is shown in Table 10. Our method achieves the best results compared to all baselines.

Table 10: BLEU scores of IWSLT translation tasks.

| Algorithm | En→De | De→En | En→Es | En→Zh | En→Fr |
|---|---|---|---|---|---|
| Standard Transformer | 28.57 | 34.64 | 39.0 | 26.3 | 35.9 |
| Feed BERT feature into embedding | 29.67 | 34.90 | 39.5 | 28.1 | 37.3 |
| Feed BERT feature into all layers of encoder | 29.61 | 34.84 | 39.9 | 28.1 | 37.4 |
| Our BERT-fused model | 30.45 | 36.11 | 41.4 | 28.2 | 38.7 |

## B.4 More baselines of IWSLT'14 German-to-English translation

We summarize the BLEU scores on IWSLT'14 De→En of existed works and our BERT-fused model approach in Table 11.

Table 11: Previous results of IWSLT'14 De→En.

| Approach | BLEU |
|---|---|
| Multi-agent dual learning (Wang et al., 2019) | 35.56 |
| Tied-Transformer (Xia et al., 2019) | 35.52 |
| Loss to teach (Wu et al., 2018) | 34.80 |
| Role-interactive layer (Weissenborn et al., 2019) | 34.74 |
| Variational attention (Deng et al., 2018) | 33.68 |
| Our BERT-fused model | 36.11 |

## B.5 COMPARISON WITH BACK TRANSLATION

When using unlabeled data to boost machine learning systems, one of the most notable approaches is back translation (briefly, BT) (Sennrich et al., 2016b): We first train a reversed translation model, use the obtained model to translate the unlabeled data in the target domain back to source domain, obtain a synthetic dataset where the source data is back-translated and finally train the forward model on the augmented dataset.

Our method has two main differences with BT method.

1. In BT, the monolingual data from the target side is leveraged. In our proposed approach, we use a BERT of the source language, which indirectly leverages the monolingual data from the source side. In this way, our approach and BT are complementary to each other. In Section 5.4, we have already verified that our method can further improve the results of standard BT on Romanian-to-English translation.

2. To use BT, we have to train a reversed translation model and then back translate the monolingual data, which is time-cost due to the decoding process. In BERT-fused model, we only need to download a pre-trained BERT model, incorporate it into our model and continue training. Besides, the BERT module is fixed during training.

On IWSLT'14, we also implement BT on wikipedia data, which is a subset of the corpus of training BERT. The model used for back translation are standard Transformer baselines introduced in Section 5, whose BLEU scores are $28.57$ and $34.64$ respectively. We back translate 1M, 2M, 5M, 15M and 25M randomly selected German sentences.

The results are reported in Table 12. The rows started with BT($\cdot$) represent the results of BT, and the numbers in the brackets are the number of sentences for back translation.

Table 12: BLEU scores IWSLT'14 En←De by BT.

| Algorithm | En→De |
|---|---|
| Standard Transformer | 28.57 |
| BERT-fused model | 30.45 |
| BT (1M) | 29.42 |
| BT (2M) | 29.76 |
| BT (5M) | 29.10 |
| BT (15M) | 28.26 |
| BT (25M) | 27.34 |

IWSLT dataset is a collection of spoken language, and the bilingual training corpus is small ($160k$). In Wikipedia, the sentences are relatively formal compared to the spoken language, which is out-of-domain of spoken languages. We can see that when using 1M or 2M monolingual data for BT, the BLEU scores can indeed improve from 28.57 to 29.42/29.76. However, simply adding more wikipedia data for BT does not result in more improvement. There is even a slight drop when adding more than 15M monolingual sentences. However, our BERT-fused model can achieve better performances than BT with wikipedia data.

## C COMPARISON OF INFERENCE TIME

Table 13: Comparisons on inference time (seconds), '+' is the increased ratio of inference time.

| Dataset | Transformer | Ours | (+) |
|---|---|---|---|
| IWSLT'14 En→De | 70 | 97 | 38.6% |
| IWSLT'14 De→En | 69 | 103 | 49.3% |
| WMT'14 En→De | 67 | 99 | 47.8% |
| WMT'14 En→Fr | 89 | 128 | 43.8% |

We compare the inference time of our approach to the baselines. The results are shown in Table 13, where from the second column to the last column, the numbers are the inference time of standard Transformer, BERT-fused model, and the increase of inference time.

Indeed, introducing BERT to encode the input brings additional inference time, resulting in about 40% to 49% increase. But considering the significant improvement of BLEU score, it is acceptable of such extra cost. We will study how to reduce inference time in the future.

## D  DOWNLOAD LINK OF PRE-TRAINED BERT MODELS

We leverage the pre-trained models provided by PyTorch-Transformers[6].

For IWSLT'14 tasks, we choose $\text{BERT}_{\text{base}}$ model with 12 layers and hidden dimension 768.

1. IWSLT14 En→{De, Es, Fr, Zh}, we choose `bert-base-uncased`.
2. IWSLT14 De→En, we choose `bert-base-german-cased`.

For WMT14 En→{Fr, De}, we choose `bert-large-uncased`, which is a $\text{BERT}_{\text{large}}$ model with 24 layers and hidden dimension 1024.

For WMT16 Ro→En, we choose `bert-base-multilingual-cased`, because there is no BERT specially trained for the Romanian.

For unsupervised En↔Fr and unsupervised En↔Ro, we choose `xlm-mlm-enfr1024` and `xlm-mlm-enro1024` respectively.

The download links are summarized as follows:

- bert-base-uncased: `https://s3.amazonaws.com/models.huggingface.co/bert/bert-base-uncased.tar.gz`.
- bert-large-uncased: `https://s3.amazonaws.com/models.huggingface.co/bert/bert-large-uncased.tar.gz`.
- bert-base-multilingual-cased: `https://s3.amazonaws.com/models.huggingface.co/bert/bert-base-multilingual-cased.tar.gz`.
- bert-base-german-cased: `https://int-deepset-models-bert.s3.eu-central-1.amazonaws.com/pytorch/bert-base-german-cased.tar.gz`.
- xlm-mlm-enfr1024: `https://s3.amazonaws.com/models.huggingface.co/bert/xlm-mlm-enfr-1024-pytorch_model.bin`.
- xlm-mlm-enro1024: `https://s3.amazonaws.com/models.huggingface.co/bert/xlm-mlm-enro-1024-pytorch_model.bin`.

## E  DETAILS OF THE NOTATIONS

Let $\text{attn}(q, K, V)$ denote the attention layer, where $q$, $K$ and $V$ indicate query, key and value respectively. Here $q$ is a $d_q$-dimensional vector ($d \in \mathbb{Z}$), $K$ and $V$ are two sets with $|K| = |V|$. Each $k_i \in K$ and $v_i \in V$ are also $d_k/d_v$-dimensional ($d_q$, $d_k$ and $d_v$ can be different) vectors, $i \in [|K|]$. The attention model works as follows:

$$\text{attn}(q, K, V) = \sum_{i=1}^{|V|} \alpha_i W_v v_i, \ \alpha_i = \frac{\exp\left((W_q q)^T (W_k k_i)\right)}{Z}, \ Z = \sum_{i=1}^{|K|} \exp((W_q q)^T (W_k k_i)), \tag{6}$$

where $W_q$, $W_k$ and $W_v$ are the parameters to be learned. In Vaswani et al. (2017), `attn` is implemented as a multi-head attention model and we omit the details here to increase readability. Following Vaswani et al. (2017), we define the non-linear transformation layer as

$$\text{FFN}(x) = W_2 \max(W_1 x + b_1, 0) + b_2, \tag{7}$$

---

[6] `https://github.com/huggingface/pytorch-transformers`

where $x$ is the input; $W_1$, $W_2$, $b_1$, $b_2$ are the parameters to be learned; $\max$ is an element-wise operator. Layer normalization is also applied following Transformer (Vaswani et al., 2017).

