# OpenReview forum: "Incorporating BERT into Neural Machine Translation"
_ICLR.cc/2020/Conference — Accept (Poster)_

### Official Review · AnonReviewer1 · 2019-10-20
**Official Blind Review #1**

**Rating:** 6

**Review:**

The paper proposes an approach to incorporate BERT pretrained sentence representations within a NMT architecture.
It shows that simply pretraining the encoder of a NMT model with BERT does not necessarily provide gains (and can even be detrimental) and proposes instead to add a new attention mechanism, both in the encoder and in the decoder. The modification is relatively simple, but provides significant improvements in supervised and unsupervised MT, although it makes the model slower and computationally more expensive. The paper contains a lot of experiments, and a detailed ablation study.

===

I'm very surprised by the results in Table 1, i.e. the fact that pretraining can decrease the performance significantly. The provided explanation "Our conjecture is that the XLM model is pre-trained on news data, which is out-of-domain for IWSLT dataset mainly about spoken languages" is not satisfactory to me. The domain mismatch is also there in the majority of GLUE tasks, SQUAD, etc. and yet pretraining with BERT significantly improves the performance on these tasks. When the encoder is pretrained with a BERT/XLM model, I assume the encoder is not frozen, but finetuned?

The description of the algorithm in Section 4 could be simplified a lot I feel. Overall, the attention in the encoder is simply replaced by two attention layers: one over the previous layer like in a standard setting, and one on top of the BERT representation. Also I don't understand why the attention over the BERT sequence is also necessary in the decoder. Shouldn't this information already be captured by the encoder output?

The Drop-Net Trick is interesting. But the fact that 1.0 gives the best performance (Section 6.2) is very unintuitive to me. This means that the model will never consider the setting with two attentions at training time, although this is what it does at test time.

In Table 6, you propose experiments with 12 and 18 layers for fair comparison, because as you mention, your model with BERT-fused has more parameters. But IWSLT is a very small dataset and it would have been surprising that using 18 layers actually helps (overfitting is much more likely in that setting). Instead, I think something like an ensemble model would be a more fair comparison. In fact, the BERT-fused is essentially an ensemble model of the encoder.
Could you try the following experiment on IWSLT, where you do not pretrain the BERT model with the BERT objective, but with a NMT encoder trained in a regular supervised setting (i.e. do not reload a BERT model, but a NMT encoder that you previously trained without the fused architecture)?

Overall, I think the gains are nice, but I would really like to see the comparison I mentioned just above, and comparisons with ensemble models. The proposed model is significantly larger / slower than the baseline models considered, and I wonder if you could not achieve the same gains with ensemble models.

Something I like about the approach is that is it quite generic in the sense that you can provide any external sequence of vectors as input to your encoder. As a result, it is possible to leverage a model pretrained with a different tokenization. Tokenization is often an issue with pretraining in NLP (how do you leverage a model trained without BPE if you actually want to use BPE in your new model). The proposed approach does not has this constraint and I think this is something you should highlight more in the paper.

===

Small details in the related work section:
- I would cite "Sutskever et al, 2014" for the LSTM encoder, along with "Hochreiter & Schmidhuber", and not only "Wu et al, 2016"
- Removing the NSP task was proposed in "Lample & Conneau, 2019", not in "Liu et al, 2019"

**Experience Assessment:**

I have published in this field for several years.

**Review Assessment: Checking Correctness Of Derivations And Theory:**

I carefully checked the derivations and theory.

**Review Assessment: Checking Correctness Of Experiments:**

I carefully checked the experiments.

**Review Assessment: Thoroughness In Paper Reading:**

I read the paper thoroughly.

---

> ### Author Response · Authors · 2019-11-15
> **Response to Reviewer#1**
>
> Thanks a lot for your valuable comments and suggestions!
>
> 1.	Yes, when the encoder is pretrained with a BERT/XLM model, it is finetuned rather than frozen.
> 2.	About Table 1 “pretraining can decrease the performance significantly”. Indeed, we have no definitive answer to explain this observation so far. Domain mismatch is one of our conjectures. Another conjecture is that XLM uses a different codebase compared to Fairseq-pytorch. We tried our best to boost the performance of XLM on IWSLT, including tuning different dropout rates and learning rates. As shown in Table 5 in the paper, for Ro->En, our reproduced Transformer baseline using Fairseq-pytorch is 33.12, while in the XLM paper, the Transformer baseline is only 28.4 (see Table 3 in XLM paper), which is a very significant gap. Similar phenomenon is also observed in WMT’14 En->De (see Appendix B1). Since Transformer baseline already achieves very high accuracy, it might be difficult for XLM to further boost the accuracy.
> 3.	We simplified the algorithm in Section 4 in the updated version. Please kindly have a check.
> About BERT-decoder attention: First, analogy to the encoder-decoder attention, we use BERT-decoder attention to allow decoder explicitly attending to the BERT output instead of leveraging this information implicitly/indirectly from encoder output. Second, we have done ablation study in Section.6-> Study for training strategy and network architecture->(3). If removing the BERT-decoder attention, the performance drops from 30.45 to 29.90, which demonstrates that leveraging BERT-decoder attention is a more effective way.
> 4.	Note that the drop-net operation is performed independently in different layers. Thus, although the self-attention and BERT-attention never meet in the same layer, they can meet across layers, e.g., self-attention in l-th layer and BERT attention in (l+1)-th layer.  Please see our code at line https://github.com/bert-nmt/bert-nmt/blob/75bd2120a0302c6ae413a58276a2c0759a19287c/fairseq/models/transformer.py#L1356 and line https://github.com/bert-nmt/bert-nmt/blob/75bd2120a0302c6ae413a58276a2c0759a19287c/fairseq/models/transformer.py#L1546.
> 5.	Following your suggestions, we replace the BERT module in our algorithm with a pretrained NMT encoder (previously trained with a different random seed and without the fused architecture). On IWSLT’14 En->De and De->En, this algorithm achieves 28.99 and 35.26 BLEU score, not as good as our method (30.45 and 36.11). This shows the advantage of BERT over a conventional encoder.
> 6.	For the ensemble results, please kindly refer to “## About better comparisons ## of Reviewer 3” and Appendix B.2.
> 7.	Thanks for your comment on advantage of the different tokenization problem in our method. As you suggested, we discussed and highlight it in the paragraph before section 4.2.
> 8.	Thanks for pointing the problems of related work. We have already corrected them.

---

### Official Review · AnonReviewer2 · 2019-10-21
**Official Blind Review #2**

**Rating:** 6

**Review:**

This paper explores the use of BERT to improve Neural Machine Translation (NMT) both in supervised, semi-supervised and unsupervised settings. The authors first show that using BERT to initialize the encoder and/or the decoder does not bring any clear improvement, while using it as a feature extractor performs better. Based on this finding, the authors propose a new approach to integrate BERT in NMT, named BERT-fused NMT, which incorporates BERT representations from the input sequence into the encoder and decoder attention mechanisms.

I am ambivalent about this paper. On the one hand, the paper presents a thorough experimental evaluation, with strong baselines (often outperforming their original implementation) and results that can be interesting from different angles, and the reported improvements are consistent. However, the paper is rather poorly written and some important details are not adequately described, which left me with some concerns and an overall negative impression as I read through the paper. More concretely:

- The paper is rather poorly written. There are many expressions that sound ungrammatical or otherwise unnatural to me (although I am not a native speaker myself) and, more importantly, the overall exposition of ideas is not sufficiently clear. I found the paper difficult to follow, and I was left with many doubts as I read through it. In addition, the style in which some results are presented is inappropriate for an academic paper (e.g. "Obviously, our proposed BERT-fused NMT can improve the BLEU scores"), although I understand that this was probably not intentional.

- To make things worse, the paper is 10 pages long, and according to the CFP reviewers are "instructed to apply a higher standard to papers in excess of 8 pages". I think that the paper could be fit in the regular 8 page limit.

- The pre-trained BERT models that the authors use were trained on different (and generally larger) training data than what they use for the NMT training (e.g. they all use Wikipedia). As such, the models that build on BERT are indirectly using this additional training data. How can we make sure that the reported improvements are not due to this additional data? What would happen if the same data was used for the baseline systems (e.g. through back-translation)? Also, please clearly state which pre-trained model you use for each specific experiment.

- The treatment of subword tokenization is not given sufficient attention and raises some concerns to me. It seems clear that the authors combine different subword tokenizations for their proposed system (i.e. BERT and the NMT encoder/decoders use a different subword vocabulary). However, it is not clear to me how this is handled in the baseline systems that use BERT for initialization only, for which a mismatch in tokenization would be problematic.

- I often find it difficult to understand what the authors did exactly for each of the reported systems. For instance, what is the difference between "Standard transformer" and "Training NMT module from scratch" in Table 6? I cannot see any yet the difference in BLEU is 1.5.

**Experience Assessment:**

I have published in this field for several years.

**Review Assessment: Checking Correctness Of Derivations And Theory:**

N/A

**Review Assessment: Checking Correctness Of Experiments:**

I assessed the sensibility of the experiments.

**Review Assessment: Thoroughness In Paper Reading:**

I read the paper at least twice and used my best judgement in assessing the paper.

---

> ### Author Response · Authors · 2019-11-15
> **Response to Reviewer#2**
>
> Thanks a lot for your valuable comments and suggestions!
>
> ## About writing ##
> We have carefully revised the paper according to your suggestions. Considering that the paper is still under review (by other reviewers), we did not compress the article within eight pages, which would lead to significant changes to the organization and correspondingly additional workload for other reviewers. We will revise it after the review period.
>
> ## About additional data ##
> Very good point! Yes, any model leveraging BERT will indirectly benefit from additional data. The difference is that back-translation (briefly, BT) leverages the unlabeled data from the target side, while we leverage the data from source side. That is, our model is complementary to BT. In Section 5.4 of the original submission, we have already verified that our method can further improve the results of BT.
> Note that it is usually costly to back translate a large amount of monolingual data due to the decoding process, and therefore BT usually takes much longer time for training. In contrast, we do not need to translate the unlabeled data when using BERT-fused model, because BERT is already pretrained and publicly available. The BERT module in our approach is fixed and does not need to be updated, which does not significantly increase training time.
>
> We back translate 1M,2M, 5M, 15M and 25M unlabeled German wiki corpus (used for training BERT) and run BT on IWSLT’14 En->De translation. The BLEU scores of above five settings are 29.42, 29.76, 29.10, 28.26 and 27.34 respectively. According to the above results, simply increasing wiki data for BT actually hurts the translation accuracy on IWSLT: 29.76 for 2M wiki data vs 27.34 for 25M wiki data. The highest BLEU score 29.76 of BT comes from 2M data, which is not as good as ours (30.45).  This verifies the effectiveness of our approach while leveraging monolingual data. Please refer to Appendix B.5 for more detailed discussions.
>
> ## The BERT model for each task ##
> We use the BERT models archived by Huggingface (https://github.com/huggingface/transformers).
> For IWSLT’14 tasks, we choose BERT_{base} models.
> 1.	IWSLT’14 En->{De, Es, Fr, Zh}, we choose ‘bert-base-uncased’.
> 2.	IWSLT’14 De->En, we choose ‘bert-base-german-cased’.
> For WMT’14 En->{Fr, De}, we choose ‘bert-large-uncased’, which is a BERT_{large} model.
> For WMT’16 Ro->En, we choose  ‘bert-base-multilingual-cased’, because there is no BERT specially trained for the Romanian.
> For the two unsupervised NMT tasks, we choose the XLM models (cross-lingual pretrained language models)
> 1.	unsupervised En->Fr, we choose ‘xlm-mlm-enfr1024’
> 2.	unsupervised En->De, we choose ‘xlm-mlm-enro1024’
> All these details are provided in Appendix D.
>
> ## Subword tokenization ##
> We assume you are talking about Table 1. While using BERT to initialize the encoder of NMT, we use BERT vocabulary and tokenization; while using XLM to initialize the encoder of NMT, we use XLM vocabulary and tokenization. We also use BERT vocabulary and tokenization for standard Transformer, which leads to similar accuracy (our 28.57 vs BERT 28.18).
>
> ## Statement ##
> We revise the ablation study considering all review comments. As stated in ``Training strategy’’ of Section 5.1, we first train a Transformer model until convergence, then use this model to initialize the encoder and decoder of the BERT-fused model. The BERT-encoder attention and BERT-decoder attention are randomly initialized. In the ablation study of Section 6, “Training NMT module from scratch” means that the encoder and decoder of BERT-fused model are not initialized from a pre-trained Transformer model but randomly initialized. We now change “Training NMT module from scratch” to “Randomly initialize the encoder/decoder of BERT-fused”.

---

### Official Review · AnonReviewer3 · 2019-10-24
**Official Blind Review #3**

**Rating:** 6

**Review:**



This paper discusses a method that effectively incorporates a (large) pre-trained LM, such as BERT and XMN, for improving the performance of NMT.

The motivation of this paper is rather straightforward and not novel; many researchers can quickly think of such an idea of incorporating the power of the recent (rapid) development of pre-training LMs into NMT.
From this perspective, this paper is not very exciting.
However, as described in the paper, we often fail to improve (or even degrade) the performance of NMT when we straightforwardly incorporate a pre-trained LM.
Thus, many researchers/developers might want to know a practical approach to integrate a pre-trained LM into NMT.
This paper provides a straightforward but smart way to incorporate pre-trained LMs, which is not trivial in the community.
In this sense, this paper might have a considerable influence on the community.
I was a bit surprised by the apparent effectiveness of the proposed method since I also have attempted to apply pre-trained LMs to NMT and have not obtained a good result.


Experimental results are mostly convincing; the authors conducted comprehensive and extensive experiments on many settings, such as supervised NMT with low- and hi-resource settings, a semi-supervised NMT setting by back-translation, document-level MT, and unsupervised NMT.
The results were also promising; the proposed method consistently outperformed conventional methods.
I think these results are useful for many readers.
Moreover, such findings also offer further insights for many researchers who aim to apply BERT to many other tasks, especially for text generation tasks.


Here are my concerns about this paper.

1, unclear explanations
The writing can be much improved. Readers might be able to guess, but several descriptions are hard to follow, or detailed explanations are missing.
For example, what is the exact operation of "function cascade"?
What is the difference between the "Training NMT module from scratch" and "Standard Transformer" in Table 6?  What is the main reason for the lower performance of (Miculicich et al. (2018)) than that of sentence-level NMT in Table 4?

2, better comparisons
I think the authors need to confirm another model setting for a fairer comparison, something like "The proposed architecture with (fixed) random vectors instead of the BERT's contextualized embeddings.
It is because we sometimes observe the improved performance for the above model comparing with the original one.
We can interpret this improvement by the effect of increasing the weight parameters for injecting the additional random vectors to the original architecture.
Therefore, I think the above model settings can improve the performance of standard Transformers, which can be a preferable counterpart of the proposed method.
Moreover, the proposed method is closely related to the model ensembling since the method utilizes two separate models.
Therefore, the authors should also report the results of model ensembling for better comparisons.

3, less discussion for the experimental results
I found minimal discussions about the results.
For example, in the ablation study, the authors only show (list) the observations of their results and no discussions.
The authors should provide discussions about how and why their method (architecture) can improve the performance compared with a similar (and current de facto standard) approach, like the fine-tuning setting that can often improve most of the other NLP tasks.



**Experience Assessment:**

I have published in this field for several years.

**Review Assessment: Checking Correctness Of Derivations And Theory:**

N/A

**Review Assessment: Checking Correctness Of Experiments:**

I carefully checked the experiments.

**Review Assessment: Thoroughness In Paper Reading:**

I read the paper thoroughly.

---

> ### Author Response · Authors · 2019-11-15
> **Response to Reviewer#3**
>
> Thanks a lot for your valuable comments and suggestions!
>
> ## About unclear explanations ##
> We have revised the paper according to your suggestions and uploaded a new version. Specifically, for your questions:
> 1.	"Function cascade" means that the functions are applied to the input in a cascaded way. We make it clearer in the current version："..., the input is processed by self-attention, encoder-decoder attention and BERT-decoder attention sequentially"  and provide a mathematical formulation. Currently, we move this part to 'Part I of Appendix B.2'.
> 2.	As stated in "Training strategy" of Section 5.1, we first train a Transformer model until convergence, then use this model to initialize the encoder and decoder of the BERT-fused model. The parameters in BERT-encoder attention and BERT-decoder attention are randomly initialized. In the ablation study of Section 6, “Training NMT module from scratch” means that the encoder and decoder of BERT-fused model are not initialized from a pre-trained Transformer model but randomly initialized. We now change “Training NMT module from scratch” to “Randomly initialize the encoder/decoder of BERT-fused” for clarity.
> 3.	Miculicich et al. (2018) released their code at https://github.com/idiap/HAN_NMT. We have already tried our best to tune this model but failed to achieve higher results than our baselines. Our conjecture is that (Miculicich et al. 2018) use a different code base (OpenNMT) instead of Fairseq-Transformer, which may cause several differences in implementation. We will conduct more study in the future.
>
> ## About better comparisons ##
> 1.	“The proposed architecture with (fixed) random vectors instead of the BERT's contextualized embedding": We implemented this algorithm and conducted experiments on IWSLT’14 En->De and IWSLT’14 De->En. Such an algorithm achieved 28.91 BLEU score for En->De and 35.00 for De->En. Indeed, this algorithm outperforms the standard Transformer, where the two BLEU scores are 28.57 and 34.64 respectively. However, its accuracy is still far-behind our proposed method (30.45 and 36.11), indicating that the improvement of our model mainly comes from pretrained BERT instead of purely increasing the number of parameters.  (See Table 6 and Section 6 -> Study for training strategy and network architecture –> (2) for more details.)
> 2.	For ensemble: On IWSLT’14 En->De, the ensemble of two, three and four standard transformer models can lead to 29.71, 30.08 and 30.18 BLEU scores respectively. Our BERT-fused model (30.45) beats all those scores.
> Furthermore, BERT-fused model can also benefit from ensemble. Ensemble of two, three and four BERT-fused models can lead to 31.09, 31.45 and 31.85 BLEU scores, outperforming the single BERT-fused model by up to 1.40 points.  Details are reported in Appendix B.2 of the updated version.
> 3.	We enriched the discussions in Section 6. Please kindly refer to the new version of our paper.

---

### Public Comment · ~SICHENG_YU1 · 2019-10-30
**Dev/Test set of IWSLT tasks**

I am confused about your split of IWSLT tasks. What do you mean by 'For other tasks, we do not lowercase the words and use the official validation/test sets of the corresponding years.' I found in IWSLT17 there is only dev2010 and several test sets. I just tried BERT in machine translation with sacreBLEU on IWSLT EN-FR using dev2010 as validation and tst2010 as test set. However I only get sacreBLEU 30.38. Did I use the wrong set split?

---

> ### Author Response · Authors · 2019-10-30
> **Re: Dev/Test set of IWSLT tasks**
>
> Hi, Sicheng.
>
> Thanks for your interests to our work.
>
> We guess that you did not find the real test set. The files you used, dev2010 and tst2010 exist in training archive. You should download the corresponding test archive in the following link:
>
> https://wit3.fbk.eu/mt.php?release=2017-01-ted-test
>
> We re-check our result of En-Fr through the following command:
>
> cat $your_output_file |  python sacreBLEU/sacrebleu.py -t iwslt17 -l en-fr
>
> and you can get
>
> BLEU+case.mixed+lang.en-fr+numrefs.1+smooth.exp+test.iwslt17+tok.13a+version.1.4.2 = 38.7 64.9/44.4/32.5/23.9 (BP = 1.000 ratio = 1.048 hyp_len = 28258 ref_len = 26962)
>
> Best,
> Authors

---

### Decision · Program_Chairs · 2019-12-19

**Decision:**

Accept (Poster)

**Comment:**

The authors propose a novel way of incorporating a large pretrained language model (BERT) into neural machine translation using an extra attention model for both the NMT encoder and decoder.   The paper presents thorough experimental design, with strong baselines and consistent positive results for supervised, semi-supervised and unsupervised experiments. The reviewers all mentioned lack of clarity in the writing and there was significant discussion with the authors. After improvements and clarifications, all reviewers agree that this paper would make a good contribution to ICLR and be of general use to the field.